# Piezo1 Channels Contribute to the Regulation of Human Atrial Fibroblast Mechanical Properties and Matrix Stiffness Sensing

**DOI:** 10.3390/cells10030663

**Published:** 2021-03-16

**Authors:** Ramona Emig, Wiebke Knodt, Mario J. Krussig, Callum M. Zgierski-Johnston, Oliver Gorka, Olaf Groß, Peter Kohl, Ursula Ravens, Rémi Peyronnet

**Affiliations:** 1Institute for Experimental Cardiovascular Medicine, University Heart Center Freiburg Bad Krozingen, and Faculty of Medicine, University of Freiburg, 79110 Freiburg, Germany; ramona.emig@universitaets-herzzentrum.de (R.E.); wiebke.knodt@gmail.com (W.K.); mkrussi@g.clemson.edu (M.J.K.); callum.michael.johnston@universitaets-herzzentrum.de (C.M.Z.-J.); peter.kohl@universitaets-herzzentrum.de (P.K.); ursula.ravens@tu-dresden.de (U.R.); 2CIBSS Centre for Integrative Biological Signalling Studies, University of Freiburg, 79104 Freiburg, Germany; 3Faculty of Biology, University of Freiburg, 79104 Freiburg, Germany; 4Institute of Neuropathology, Medical Center-University of Freiburg, Faculty of Medicine, University of Freiburg, 79106 Freiburg, Germany; oliver.gorka@uniklinik-freiburg.de (O.G.); olaf.gross@uniklinik-freiburg.de (O.G.); 5Signalling Research Centres BIOSS and CIBSS, University of Freiburg, 79104 Freiburg, Germany; 6Center for Basics in NeuroModulation (NeuroModulBasics), Faculty of Medicine, University of Freiburg, 79104 Freiburg, Germany

**Keywords:** heart, cardiac fibrosis, integrin, actin, cytoskeleton, adhesion, Young’s modulus, calpain, ROCK, FAK

## Abstract

The mechanical environment of cardiac cells changes continuously and undergoes major alterations during diseases. Most cardiac diseases, including atrial fibrillation, are accompanied by fibrosis which can impair both electrical and mechanical function of the heart. A key characteristic of fibrotic tissue is excessive accumulation of extracellular matrix, leading to increased tissue stiffness. Cells are known to respond to changes in their mechanical environment, but the molecular mechanisms underlying this ability are incompletely understood. We used cell culture systems and hydrogels with tunable stiffness, combined with advanced biophysical and imaging techniques, to elucidate the roles of the stretch-activated channel Piezo1 in human atrial fibroblast mechano-sensing. Changing the expression level of Piezo1 revealed that this mechano-sensor contributes to the organization of the cytoskeleton, affecting mechanical properties of human embryonic kidney cells and human atrial fibroblasts. Our results suggest that this response is independent of Piezo1-mediated ion conduction at the plasma membrane, and mediated in part by components of the integrin pathway. Further, we show that Piezo1 is instrumental for fibroblast adaptation to changes in matrix stiffness, and that Piezo1-induced cell stiffening is transmitted in a paracrine manner to other cells by a signaling mechanism requiring interleukin-6. Piezo1 may be a new candidate for targeted interference with cardiac fibroblast function.

## 1. Introduction

Fibroblasts sense and adapt to changes in their mechanical environment. The mechanical properties of the extracellular matrix (ECM) in particular, act as a driver for a number of cell functions including differentiation, motility, myofibroblast phenoconversion, and collagen production [1]. Mechanical properties of the ECM can change drastically during physiological (e.g., development) and pathophysiological conditions (e.g., mechanical overload) in a number of organs [2], including the heart. For example, the early inflammatory phase following myocardial infarction is characterized by a softening of the myocardium down to stiffnesses of a few kilopascals [3], due to collagen degradation and cardiomyocyte death. In contrast, values exceeding 50 kilopascals have been observed in fully mature ventricular scars several months or years after injury [4,5,6,7]. The current state of knowledge about the sensing of passive mechanics from cell to tissue levels in the heart has been reviewed recently in detail [8]. The molecular mechanisms underlying cell adaptation to matrix stiffness are still ill-understood.

The main cellular components that mediate the sensing and regulation of ECM mechanics are transmembrane receptors of the integrin family, proteins associated with focal adhesions, and the actomyosin cytoskeleton [9]. Integrins are major adhesion receptors of the cell [10]. By physically linking the ECM to the cytoskeleton, they transmit forces and deformation between the inside of the cell and the ECM [11].

Integrin signaling involves a large number of proteins, including talin and various kinases. Talin is instrumental for cell adhesion by linking integrins to the cytoskeleton. Integration of internal and external stimuli allows talin to activate integrins (i.e., convert them to high-affinity states so they can bind their substrate) “on demand” [12]. Kinases play a central role in the transmission of signals from focal adhesions to the inside of a cell. In particular the focal adhesion kinase (FAK) is recruited to focal adhesions and then activated in response to integrin-β1 activation. This is important for adhesion turnover, Rho family guanosine triphosphatase (GTPase) activation, cell migration, and crosstalk with signals from other receptors (e.g., growth factors, [13]). The Rho family of small GTPases and their downstream effectors such as Rho-associated protein kinase (ROCK) and myosin light chain kinases are major players in the assembly of the focal adhesion complex linking ECM to the cytoskeleton. For more detail on integrin signaling please see previous communications [14,15,16].

Integrin signaling interacts with numerous additional pathways. Recently, crosstalk between mechano-sensitive ion channels and integrin signaling has been proposed. The canonical transient receptor potential channel 6 (TRPC6) binds to and activates calpain, independently of its activity as an ion channel, and regulates podocyte cytoskeleton organization, cell adhesion, and motility of podocytes [17]. Piezo1, another cation non-selective stretch-activated channel (SAC) involved in mechano-transduction [18,19], contributes to a number of physiological and pathophysiological processes, as reviewed elsewhere [20,21,22,23]. Piezo1 is present in cardiac fibroblasts [24] but its contribution to the functions of this cell type remains to be explored.

Piezo1 is a large homotrimer with more than 2500 amino acids, including 38 transmembrane helices per monomer [25]. It is widely distributed throughout different species and cell types, and it has been reported in focal adhesions (for example in Chinese hamster ovary cells, *Drosophila* glioblastoma stem cells [26], and human neural stem cells [27]). Similar to integrins, activation of Piezo1 can be altered by stimuli from the inside or the outside of a cell. In human neuronal stem cells, the actomyosin cytoskeleton generates sufficient forces via myosin II phosphorylation to open Piezo1 and generate Ca^2+^ flickers at focal adhesions [28]. Piezo1 activity at focal adhesions has been shown to activate integrin–FAK signaling in glioblastoma and neural stem cells via Ca^2+^-mediated signaling [26,28,29]. A potential mechanism for Piezo1-mediated integrin activation has been explored in Chinese hamster ovary cells, where recruitment of the small GTPase RRas to the endoplasmic reticulum is necessary to activate the Ca^2+^-activated protease calpain, increasing Ca^2+^ release from cytoplasmic stores [29]. In addition, it was shown that Piezo1 is sensitized to pulling forces from the outside by binding to collagen VI in human neuroblastoma cells [30]. Further connecting Piezo1 with outside-in signaling, channel expression has been shown to increase (≈1.4 fold) in stem cells cultured on stiffer polyacrylamide gels (5 kPa vs. 0.1 kPa, [26]). Along similar lines, recruitment of Piezo1-expressing monocytes (required for vascularization of implanted, hydrogel-based cardiac tissue patches) depends on physiological hydrogel stiffness in mice [31].

Piezo1 expression is altered in a number of diseases, for example in amyloid-responsive cells in Alzheimer’s disease [32] or in red blood cells in hereditary xerocytosis [33]. Piezo1 upregulation contributes to stiffening of aggressive gliomas in *Drosophila* [26]. Furthermore, Piezo1 expression in human atrial fibroblasts was reported to contribute to enhanced secretion of interleukin-6 (IL-6), a profibrotic cytokine [24]. In mouse cardiac fibroblasts, it was shown that the secretome is modulated by pro-fibrotic stimuli, including stiff growth matrices and transforming growth factor β [34]. Our own data suggest that Piezo1 expression and activity are increased in fibroblasts in the context of atrial fibrillation (AF, [35]). To explore the role of Piezo1 in the control of cell mechanical properties and cell adaptation to changes of matrix stiffness, the present study combines human cell culture systems and hydrogels of different stiffness with nanoindentation and imaging. Our results demonstrate that Piezo1 expression contributes to: (i) cytoskeleton organization, (ii) cell mechanical properties, and (iii) cellular adaptation to changes in matrix stiffness. These effects can be transmitted to other cells via secreted IL-6.

## 2. Materials and Methods

### 2.1. Cell Culture

#### 2.1.1. Cell Types and Maintenance

Human embryonic kidney cells (HEK 293T/17, ATCC-LGC Standards, Manassas, Virginia, USA) were cultured in Dulbecco’s Modified Eagle Medium (DMEM) with low glucose (D6046, Sigma-Aldrich, Hamburg, Germany) supplemented with 10% fetal calf serum (F9665, Sigma-Aldrich, Hamburg, Germany) and 1% penicillin/streptomycin (P4333, Sigma-Aldrich, Hamburg, Germany). HEK cells offer the advantage of being easy to transfect even with large constructs and thus represent a widely used cell culture model for overexpression experiments. We use HEK cells to overexpress the 2521 amino acid protein Piezo1 (Uniprot entry Q92508, Section 2.1.2) and to test the effects of various compounds on Piezo1-induced cell stiffening (Section 2.1.3).

A human atrial fibroblast line (HAF, [36]) was cultured in DMEM supplemented with 2 mM L-alanyl-L-glutamine (GlutaMAX, 31966021, LifeTechnologies, Darmstadt, Germany), 10% fetal calf serum and 1% penicillin/streptomycin. At ≈90% confluence, cells were detached using Trypsin-ethylene-diamine-tetraacetic acid (59418C, Sigma-Aldrich, Hamburg, Germany) and seeded in fresh polystyrene flasks (Z707538, TPP, Trasadingen, Switzerland) for maintenance, or on various substrates for experiments (see below).

#### 2.1.2. Transfection

For transient overexpression of enhanced green fluorescent protein (EGFP), of EGFP and Piezo1, or of EGFP and TRPC6, cells were transfected with the respective plasmids (vector backbone: pIRES2_EGFP, 6029-1, Addgene, Watertown, MA, USA) 24 h after seeding, using JetPEI transfection reagent (101-10N, Polyplus transfections, Illkirch, France) according to manufacturer’s instructions and as described previously [37]. For each 35 mm diameter well, 1 µL JetPEI transfection reagent and 0.5 µg of plasmid DNA were mixed in 100 µL NaCl solution (150 mM) and incubated for 20 min at room temperature (≈21 °C) before being added to the wells. Successfully transfected cells were identified by cytosolic EGFP fluorescence. Cells transfected with Piezo1 and EGFP will further be referred to as “Piezo1”, cells with TRPC6 and EGFP as “TRPC6”, and EGFP-expressing control cells with neither Piezo1 nor TRPC6 as “EGFP”.

Knock-down of target genes was performed using previously validated SMARTpool siRNA (L-020870-03 against Piezo1 (siPiezo1), D-001810-10-05 as a non-targeting control (siNT), Horizon Discovery, Cambridge, United Kingdom [24]). The day after cell seeding, HAF were transfected with the respective siRNA using HiPerFect (301704, Qiagen, Hilden, Germany). For each 35 mm diameter well, 12 µL of transfection reagent and 8 nM of siRNA were diluted in 187.2 µL of DMEM + GlutaMAX in 15 mL tubes. After 15 min of incubation at room temperature, the transfection mix was filled up to 2 mL with complete culture medium (DMEM + GlutaMAX + 10% fetal calf serum + 1% penicillin/streptomycin) and carefully mixed. The old culture medium was removed from the cells and replaced by the transfection mix.

#### 2.1.3. Compounds and Treatments

The compounds listed in Table 1 (or the respective solvents for controls) were added to the culture medium 24 h after transfection. Functional experiments were performed 2 days after exposure to compounds (i.e., 3 days post-transfection).

Unless explicitly stated otherwise, cells were cultured in the presence of antibiotics (172 µM streptomycin and 200 µM penicillin). Streptomycin is a hydrophilic, and thus non-cell-permeable [38,39], blocker of cation non-selective SAC, including Piezo1. For experiments without streptomycin, the culture medium was replaced by medium without antibiotics (DMEM, low glucose with 10% fetal calf serum) 24 h after transfection. Piezo1 has also been detected in the endoplasmic reticulum membrane [29], which is not accessible to hydrophilic streptomycin. As the endoplasmic reticulum constitutes a major reservoir for Ca^2+^, we further assessed the role of the Ca^2+^-activated protease calpain by directly blocking it, using the inhibitory peptide N-acetyl-leucin-leucin-norleucinal (ALLN, [40]). Cell morphology and detachment in response to ALLN treatment were assessed by light microscopy (10× magnification, Nikon Eclipse TS100 inverted microscope equipped with Leica EC3 camera). Cell detachment was quantified as percentage of rounded-up cells present on the bottom of the dish in a field of view.

To interfere directly with integrin-β1 signaling, conformation-specific antibodies, stabilizing either the active (P5D2) or inactive (HMβ1.1) conformation of integrin-β1 were used [41]. Further, integrin-β1 downstream effectors FAK and ROCK were inhibited using the small molecules PF-00562271 and Y-27632, respectively. A monoclonal antibody was used to neutralize secreted IL-6 in the culture medium.

### 2.2. Cell Culture Matrices with Different Stiffness

Hydrogels with light-tunable mechanical properties (CyPhyGels) were prepared as described previously [42,43]. In short, the cyanobacterial photoreceptor-1 was recombinantly expressed, purified and covalently coupled to 8-arm polyethylene glycol. For casting, 30 µL of CyPhyGel solution was spread on square (22 by 22 mm) coverslips, resulting in CyPhyGels with ≈100 µm thickness, tunable by illumination with light of different wavelengths to a stiffness between 2.7 and 4.6 kPa [42,44].

For cell experiments, coverslips with CyPhyGels were placed into plastic culture dishes with 35 mm diameter, HAF were seeded on top at a density of 2750 cells/cm^2^ and transfected 24 h after seeding (for detail see description of individual experiments). All experiments using CyPhyGels were performed under green light illumination to prevent unintended changes in mechanical properties of the culture substrate.

### 2.3. Nanoindentation

For nanoindentation experiments, HEK cells were seeded at a density of 5500 cells/cm^2^ and HAF at 2750 cells/cm^2^ on CyPhyGels (as described above) or in plastic tissue culture dishes with 35 mm diameter (93040, TPP; 2 mL of culture medium/dish) and transfected and/or treated as indicated for individual experiments. Nanoindentation was performed 3 days post-transfection. Before nanoindentation experiments, the culture medium was replaced by phosphate buffered saline (containing (in mM): NaCl 137, KCl 2.7, Na_2_HPO_4_ 10, and KH_2_PO_4_ 1.8; pH 7.4, 300 mOsm/L) at room temperature and experiments were performed within 1 h of taking cells out of the incubator.

Nanoindentation experiments were performed using the Chiaro system (Optics11, Amsterdam, Netherlands). Optical and geometrical calibrations were performed according to the manufacturer’s instructions. Spherical glass tips with 3.0–3.4 µm radius, attached to cantilevers with a spring constant between 0.012 N/m and 0.030 N/m were used to indent cells (Figure 1A). The cell surface was identified manually by approaching it in 1 µm steps. After touching the cell, the tip was lifted by 5 µm and a displacement of 10 µm was initiated with a speed of 5 µm/s [36]. The probe was then held at maximal motor displacement for 2 s before being retracted at the same speed (Figure 1B). Each cell was indented once at 3 different positions (excluding the region containing the nucleus). The effective Young’s modulus (E_eff_) was derived from force vs. indentation curves, using a Hertzian model for contact mechanics (Figure 1C, [45]) under the assumption of a Poisson ratio of 0.5 which is customarily used for mechanical testing of cells [46]. The height of HEK cells was 20.0 ± 0.9 µm (n = 16), that of HAF was 9.2 ± 0.4 µm (n = 14). As the Hertzian model is only valid for deformations up to 10% of sample thickness [47], no more than the first 2 µm of indentation were considered for HEK cells and no more than 0.9 µm for HAF. For such small deformations, the body of mildly-structured cells (such as cultured HEK or HAF) is usually considered homogeneous [48,49,50].

### 2.4. Cytoskeleton: Staining, Image Acquisition, and Data Analysis

For imaging of the actin cytoskeleton, HAF were seeded onto borosilicate glass coverslips (#1.5, 631-0151, VWR, Ismaning, Germany) inside 24-well plates (662160, Greiner Bio-One, Frickenhausen, Germany) at a density of 1500 cells/cm^2^ (0.5 mL of culture medium/well), followed by transfection 24 h later. Then, 3 days post-transfection, cells were chemically fixed using a 4% para-formaldehyde solution. F-actin was stained using Phalloidin-iFluor-647 (ab176759, Abcam, Berlin, Germany), nuclear counterstain was performed using 4′,6-diamidin-2-phenylindol (DAPI, D1306, ThermoFisher, Dreieich, Germany), and coverslips were mounted using PermaFluor mounting medium (60085968, ThermoFisher, Dreieich, Germany).

Imaging was performed on a Leica TCS SP8 X laser scanning confocal microscope using a 63× glycerol immersion objective with a numerical aperture of 1.3. Z-stacks with a step size of 1 µm covering the whole height of a single cell were acquired with an optical thickness of 0.33 µm/plane. For analysis, all planes were background subtracted using the corresponding function in Fiji [51], and maximal intensities were projected onto a single plane. Average fluorescence intensity and area occupied by each cell were calculated after manually outlining cell borders based on the F-actin image. Spatial organization and apparent thickness of actin bundles were analyzed spectrally using Cytospectre [52]. In fluorescent microscopy, fine structures appear broadened due to diffraction. The size of the point spread function is in the order of half the wavelength. Assessing differences in spatial extent below this range can be problematic but larger changes can be easily detected. The apparent bundle thickness is a reliable parameter for characterizing cytoskeletal reorganization in the microscale. However, due to the limited resolution it is not possible to obtain detailed information on nano-structural rearrangements. Actin bundle orientation parameters used in this study are angular standard deviation (SD), which describes the variation of actin bundle orientations, and circular kurtosis, a measure for the peakedness of the distribution of orientations.

### 2.5. Gene Expression Analysis

Three days post-transfection, isolation of mRNA from cultured HAF was performed using a commercial RNA isolation kit (RNeasy Micro Kit, Cat. no.: 74004). In brief, HAF were washed once with phosphate buffered saline and subsequently overlaid with “RLT” lysis buffer (40 µL/cm^2^) supplemented with 2-mercaptoethanol (1:100). After incubation for 10 min at room temperature, cell lysates were scraped off the plates and transferred to 1.5 mL reaction tubes. For CyPhyGels, cell lysate and CyPhyGel were scraped off and transferred to 1.5 mL reaction tubes together. The lysate was then cleared from CyPhyGel by centrifugation (5 min at 5000× *g*). RNA isolation was performed according to manufacturer’s instructions. RNA concentration was determined spectrometrically. Per sample, 100 ng of RNA were utilized to generate complementary DNA (cDNA) using TaqMan Reverse Transcription Reagents (N8080234, ThermoFisher, Dreieich, Germany).

Relative mRNA expression levels were determined by qPCR. cDNA was amplified in TaqMan Fast Advanced Master Mix (4444556, ThermoFisher, Dreieich, Germany) for a total of 40 cycles, using the assays listed in Table 2. Levels of mRNA expression of Piezo1 and Piezo2 were normalized to the expression level of glyceraldehyde-3-phosphate dehydrogenase as internal reference.

### 2.6. IL-6 Measurements by Enzyme-Linked Immunosorbent Assay

Cell-free supernatants from HAF, transfected with either the EGFP or Piezo1 expression plasmid, were collected just before the start of nanoindentation experiments (3 days post-transfection) and stored at −20 °C. IL-6 was detected (in Nunc MaxiSorp 96-well plates (11530627, Invitrogen, Karlsruhe, Germany)) using a colorimetric assay based on the IL-6 Human Uncoated ELISA Kit (88-7066-88, Invitrogen) according to manufacturer’s instructions. Absorbance was measured with a Microplate Reader (Tecan Infinite 200) and analyzed with Magellan data analysis software.

### 2.7. Piezo1 Protein Level Detected by Western Blot

Three days after transfecting HEK cells and HAF with the respective plasmids or siRNAs, cells were lysed in radio immunoprecipitation buffer (containing (in mM): Tris-HCl 50 and NaCl 150; and, (in % *w/v*) NP-40 1, sodium deoxycholate 1, and sodium dodecyl sulphate 0.1) supplemented with protease inhibitors (1:200, 539134, Calbiochem, Darmstadt, Germany) for 15 min on ice, followed by centrifugation (15 min at 15,000× *g*, 4 °C). Subsequently, cleared lysates (supernatants) were subjected to sodium dodecyl sulphate polyacrylamide gel electrophoresis using 7% polyacrylamide gels and transferred to nitrocellulose membranes. These were saturated by incubation in 5% bovine serum albumin in phosphate buffered saline for 1 h at room temperature. Primary antibodies against Piezo1 (generated in rabbit, 15939-1-AP, Proteintech, Manchester, United Kingdom) and glyceraldehyde-3-phosphate dehydrogenase (generated in mouse, G8795, Sigma-Aldrich, Hamburg, Germany) were applied 1:1000 in 5% bovine serum albumin overnight at 4 °C with gentle agitation. Horseradish peroxidase-coupled secondary antibodies (anti-rabbit, 7074S, Cell Signaling Technologies, Danvers, MA, USA, and anti-mouse, HAF007, R&D Systems, Minneapolis, MN, USA) were applied 1:5000. Membranes were visualized using SuperSignal West Pico PLUS Chemiluminescence Substrate or Femto Maximum Sensitivity Substrate (34580 or 34096, ThermoFisher) and recorded using a Fusion-Fx gel documentation system (Vilber, Eberhardzell, Germany).

### 2.8. Patch-Clamp Recording of Piezo1 Activity

The patch-clamp technique was used to evaluate the presence of functional Piezo1 at the plasma membrane. Cell-attached patch-clamp recordings were performed using bath and pipette solutions previously described for characterizing Piezo1 channels [37]. The pipette medium contained (in mM): NaCl 150, KCl 5, CaCl_2_ 2, and HEPES 10; pH 7.4 with NaOH, 310 mOsm/L and the bath medium contained (in mM): KCl 155, EGTA 5, MgCl_2_ 3, and HEPES 10; pH 7.2 with KOH, 310 mOsm/L. Pressure pulses of increasing amplitude (from 0 to −60 mmHg, in −10 mmHg increments) were applied through the recording electrode using a pressure-clamp device (ALA High Speed Pressure Clamp-1 system; ALA Scientific, Farmingdale, NY, USA). Experiments were performed at room temperature (20 °C), using a patch-clamp amplifier (200B, Axon Instruments, San Jose, CA, USA) and a Digidata 1440A interface (Axon Instruments). Recorded currents were digitized at 3 kHz, low-pass filtered at 1 kHz, and analyzed with pCLAMP10.3 software (Axon Instruments) and OriginPro 2019 (OriginLabCorporation, Northampton, MA, USA).

### 2.9. Statistical Analyses

Individual data points deviating by more than 3 standard deviations from the mean of the raw data set were defined as outliers and removed from further analysis (in total 1.69% of cells). Normal distribution of the data was assessed for each group using Shapiro–Wilk test and had to be rejected for most groups. Thus, to determine statistical significance of differences between experimental groups, non-parametric Kruskal–Wallis–ANOVA has been performed, followed by post-hoc Dunn’s comparison of the means. Groups were considered significantly different with a *p*-value <0.05. Data is presented as mean ± standard error to the mean, and as single data points. All statistical analyses were performed in OriginPro 2019 (OriginLabCorporation). Graphical summary of the main results was created with BioRender.com.

With exception of Figure 4, all experiments on HAF reported in this study were performed side-by-side with controls (i.e., all conditions in an individual figure panel were tested on the same day) to account for variations between experiments.

## 3. Results

### 3.1. Piezo1 Overexpression in HEK Cells Leads to Increased Cell Stiffness

To mimic Piezo1 upregulation, as seen in fibroblasts from AF patients, we overexpressed Piezo1 in HEK cells. Using nanoindentation, we determined the stiffness of cells overexpressing EGFP or Piezo1 (Figure 2A). Average stiffness of Piezo1-overexpressing cells was 1.7 times higher than that of EGFP control cells (n = 252 cells with EGFP, n = 247 with Piezo1, from N = 17 experiments; Figure 2B). Piezo1 overexpression was confirmed at functional (patch-clamp) and biochemical (Western blot) levels (Appendix A). Neither the stiffness of EGFP control cells, nor that of Piezo1-overexpressing cells was affected by the solvents used in this study (0.1% dimethyl sulfoxide (DMSO), or 0.09% Na-azide; Appendix A). Piezo1-induced cell stiffening will be abbreviated PiCS throughout the manuscript. In response to overexpression of TRPC6, another cation non-selective SAC [53], cell stiffness was not significantly different from EGFP control cells (Figure 2C). This suggests that the observed effect on cell stiffness is not a general response to cation non-selective SAC overexpression.

### 3.2. PiCS Does Not Require Ion Flux Through the Channel, Nor Calpain Activity in HEK Cells

We assessed cell stiffness in Piezo1-overexpressing HEK cells in the presence and absence of antibiotics, including the SAC blocker streptomycin. We found that PiCS occurred in presence and absence of antibiotics (1.7-fold PiCS with antibiotics (n = 42 with EGFP, n = 40 with Piezo1, N = 3), vs. 1.6-fold PiCS without antibiotics (n = 44 with EGFP, n = 46 with Piezo1, N = 3), Figure 3A). This led to the conclusion that the function of Piezo1 as an ion-conducting channel at the plasma membrane is not required for PiCS.

Inhibition of the Ca^2+^-activated protease calpain by treating HEK cells with 5 or 10 µM ALLN for 48 h did not abolish PiCS, observed in solvent-treated control cells (2.1-fold PiCS with DMSO (n = 17 with EGFP, n = 20 with Piezo1, N = 2), vs. 2.2-fold PiCS with 5 µM ALLN (n = 30 with EGFP, n = 30 with Piezo1, N = 3), or 2.4-fold PiCS with 10 µM ALLN (n = 18 with EGFP, n = 29 with Piezo1, N = 3), Figure 3B). ALLN-treated cells (10 µM), whether expressing EGFP or Piezo1, had a different morphology and showed significantly increased cell detachment (Appendix A).

### 3.3. PiCS in HEK Cells Requires Components of the Integrin Signaling Pathway

To assess whether PiCS involves intracellular signal transduction from integrin-β1, we used conformation-specific antibodies to activate or inactivate integrin-β1. We found that constitutive activation of integrin-β1 signaling by P5D2 significantly increased stiffness in EGFP-expressing cells, compared to solvent-treated EGFP cells (1.7-fold difference (n = 31, N = 3 with Na-azide, n = 46, N = 4 with P5D2)). There was no significant further increase in cell stiffness upon Piezo1 overexpression in P5D2 exposed cells (n = 31, N = 3 with Na-azide, n = 51, N = 4 with P5D2). In contrast, blocking integrin-β1 activation by HMβ1.1 had no significant effect on cell stiffness compared to vehicle-treated cells (n = 31, N = 3 with Na-azide, n = 55, N = 3 with HMβ1.1), and it did not prevent PiCS (1.9-fold PiCS with Na-azide (n = 31 with EGFP, n = 31 with Piezo1, N = 3), 1.6-fold PiCS with HMβ1.1 (n = 55 with EGFP, n = 51 with Piezo1, N = 3, Figure 3C). Thus, we assume that integrin-β1 is not directly involved in PiCS.

Next, the contribution of known downstream effectors of integrin-β1 signaling, FAK and ROCK, was assessed. Inhibition of FAK using the small molecule inhibitor PF-00562271 abolished PiCS, without significant effect on EGFP control cells. In contrast, when ROCK was inhibited by Y-27632, the stiffness of both EGFP and Piezo1-expressing cells was reduced compared to DMSO controls independently from Piezo1 expression levels, while PiCS was conserved (1.6-fold PiCS with DMSO vehicle (n = 96 with EGFP, n = 94 with Piezo1, N = 6), vs. no PiCS with PF-00562271 (n = 57 with EGFP, n = 58 with Piezo1, N = 3), vs. 1.5-fold PiCS with Y-27632 (n = 57 with EGFP, n = 53 with Piezo1, N = 3), Figure 3D).

### 3.4. PiCS Is Conserved in HAF

Based on insight into the mechanisms of PiCS in HEK cells, we aimed to translate our findings to HAF, since we found Piezo1 expression to be upregulated in primary atrial fibroblasts from patients in AF [35]. We assessed cell stiffness in HAF in response to Piezo1 overexpression or knock-down. We first observed that cells exposed to siNT as controls for Piezo1 knock-down experiments were 1.8-fold stiffer than untreated EGFP-expressing cells used as controls in Piezo1 overexpression experiments (Figure 4A). The reasons for this difference are not clear. Data were thus normalized in subsequent analyses to their respective control group (EGFP for Piezo1 overexpression, siNT for Piezo1 knock-down, Figure 4B).

Overexpression of Piezo1 led to a pronounced (2.7-fold) increase in HAF stiffness compared to EGFP control cells (n = 24 with EGFP, n = 24 with Piezo1, N = 2), while siRNA-mediated Piezo1 knock-down reduced HAF stiffness to 0.6 of siNT control cells (n = 89 with siNT, n = 83 with siPiezo1, N = 3, Figure 4B), confirming a role for Piezo1 in setting cell stiffness in HAF.

### 3.5. Piezo1 Expression Affects Architecture of the Actin Cytoskeleton in HAF

Cellular mechanical properties are related to the composition and organization of their cytoskeleton. We used fluorescently-labelled phalloidin to stain the F-actin network in HAF and assessed effects of overexpression or knock-down of Piezo1 (Figure 5A). We noted a significant difference in average fluorescence intensity between the two control groups (1.2 times higher in siNT vs. EGFP cells (n = 76 with EGFP, n = 49 with siNT, N = 3), Appendix A), in line with stiffness recordings (above). We, thus, normalized data on Piezo1 overexpression or knock-down to their respective controls. Absolute values for all parameters can be found in Appendix A.

Average fluorescence intensity was 1.6 times higher in Piezo1-overexpressing HAF compared to EGFP control cells (n = 76 with EGFP, n = 74 with Piezo1, N = 3), while Piezo1 knock-down resulted in a reduction to 0.7 times the fluorescence intensity of siNT control cells (n = 49 for siNT, n = 49 for siPiezo1, N = 3, Figure 5B left). Piezo1 overexpression resulted in a small but significant increase in the area of a cell that was covered by actin bundles compared to EGFP control cells, while Piezo1 knock-down did not significantly affect the area covered by actin bundles compared to siNT control cells (Figure 5B middle). No changes in total cell area were identified (Figure 5B right). To gain further insight into the microscale architecture of the actin cytoskeleton, we analyzed average apparent thickness of actin bundles and their orientation. In HAF overexpressing Piezo1, we found actin bundles to be significantly thicker than those in EGFP cells, while Piezo1 knock-down did not significantly affect apparent actin bundle thickness compared to siNT controls (1.2 times thicker bundles with Piezo1 overexpression vs. no discernible difference with Piezo1 knock-down, Figure 5C left).

In response to Piezo1 overexpression, angular SD of actin bundle orientation did not differ significantly from control cells, while Piezo1 knock-down resulted in a significantly higher angular SD, indicative of a less-ordered actin network (Figure 5C middle). Circular kurtosis was significantly higher in response to Piezo1 overexpression, but not affected by Piezo1 knock-down (Figure 5C right). Normalized data suggest that Piezo1 increases the thickness of actin bundles and favors their more ordered arrangement.

### 3.6. Piezo1 Is Required for Cell Stiffness Adaptation to Matrix Stiffness in HAF

After culturing HAF on stiff or soft CyPhyGels for 4 days, we found that cell stiffness adapts to CyPhyGel stiffness (1.6-fold higher HAF stiffness on stiff compared to soft CyPhyGel, n = 30 on stiff, n = 35 on soft, N = 3; Figure 6A), results in line with previous observations [54]. This adaptation was abolished by Piezo1 knock-down (n = 32 on stiff and n = 31 on soft, N = 3, Figure 6A). While Piezo1 knock-down upon siRNA exposure was confirmed, Piezo1 mRNA level did not differ significantly between HAF on stiff and soft CyPhyGels within control or knock-down groups (n = 6 dishes per condition, N = 3, Figure 6B). As compensatory mechanisms between Piezo1 and Piezo2 have been described [55] and because of the role of Piezo2 in matrix stiffness sensing [56], we additionally analyzed its mRNA level. The expression level of Piezo2 was not significantly affected by CyPhyGel stiffness or Piezo1 knock-down (Figure 6C).

### 3.7. PiCS Is Transmitted to Neighboring HAF by IL-6

As shown before, the stiffness of Piezo1-overexpressing HAF is higher than that of EGFP-expressing control cells (2.1-fold PiCS, n = 59 with EGFP, n = 63 with Piezo1, N = 5). Surprisingly though, non-transfected (non-fluorescent) HAF cells *in the same dish* (i.e., cells neighboring to those in whom Piezo1 was overexpressed), was also significantly higher than that of non-transfected HAF in EGFP dishes (1.6-fold higher, n = 35 with EGFP, n = 36 with Piezo1, N = 3). This suggests that Piezo1-overexpressing cells may affect the stiffness of non-transfected HAF in the same dish (Figure 7A).

To assess whether IL-6 might mediate paracrine effects between transfected and non-transfected cells in the same dish, we compared IL-6 levels in the culture medium of EGFP- and Piezo1-transfected dishes. Enzyme-linked immunosorbent assay confirmed a 1.4-fold higher IL-6 concentration in the culture medium of Piezo1-transfected cells, compared to EGFP controls (n = 8 dishes with EGFP, n = 9 dishes with Piezo1, N = 3; Figure 7B).

Finally, mechanical properties of non-fluorescent (non-transfected) cells in dishes that contained HAF expressing either EGFP or Piezo1 were assessed in the presence or absence of an IL-6 neutralizing monoclonal antibody. In the presence of a control IgG that does not interfere with IL-6, non-fluorescent cells in Piezo1 dishes showed a 1.4-fold PiCS compared to non-fluorescent cells in EGFP control dishes (n = 51 with EGFP, n = 50 with Piezo1, N = 3). Application of the IL-6-neutralizing antibody abolished paracrine induction of PiCS, without significantly affecting stiffness of EGFP control cells (n = 49 with EGFP, n = 51 with Piezo1, N = 3; Figure 7C). Additionally, while application of a control IgG had no significant effect on PiCS in fluorescent (transfected) cells (1.9-fold PiCS, n = 28 with EGFP, n = 28 with Piezo1, N = 3), PiCS was abolished by neutralizing IL-6 even in fluorescent cells (n = 56 with EGFP, n = 53 with Piezo1, N = 3; Figure 7D). We conclude from this data that PiCS requires an increase of IL-6 secretion, which then mediates both, the stiffness increase of transfected cells in an autocrine, and that of non-transfected cells in a paracrine manner.

## 4. Discussion

This study provides evidence for a contribution of Piezo1 to the regulation of mechanical properties in HEK cells and HAF. Overall, our data suggest that (a) Piezo1 expression, but not its activity as an ion channel at the plasma membrane, is required for PiCS, (b) PiCS involves components of canonical integrin mechano-signaling, (c) PiCS is mediated by an autocrine mechanism that requires an increase in IL-6 secretion, which can additionally transmit PiCS to neighboring cells in a paracrine manner, and (d) Piezo1 is an essential component of cell stiffness sensing. Changes of cell mechanical properties in response to varying Piezo1 expression levels may be attributed to changes in the organization of the actin cytoskeleton. Figure 8 summarizes our conclusion from both cell types.

### 4.1. Mechanism Underlying PiCS

To obtain insight on signaling mechanisms required for PiCS, we used HEK cells, a routinely employed cell culture model. Their main advantage for our study is that they can be easily transfected with large constructs like Piezo1 (2521 amino acids, Uniprot entry Q92508) while endogenous Piezo1 expression level is low. Publications connecting Piezo1 to mechano-signaling mediated by integrins are based mainly on Piezo1-mediated elevation of intracellular Ca^2+^, entering the cytosol either from the extracellular space [57] or from the endoplasmic reticulum [29]. As PiCS is not suppressed in our hands by blocking Piezo1 ion channel activity using antibiotics, including streptomycin, it is unlikely that Ca^2+^ influx via the plasma membrane is required for PiCS and matrix stiffness sensing. This is similar to prior reports on crosstalk between the mechano-sensitive channel TRPC6 and integrin signaling in podocytes, which also does not rely on ion flux through the channel. While TRCP6 requires calpain to regulate podocytes adhesion and cytoskeleton [17], our data on Piezo1 in HEK cells suggest a mechanism that is independent from calpain.

While our data suggest that PiCS does not require Piezo1-mediated influx of extracellular Ca^2+^, further studies, such as using a non-conductive or a truncated mutant of Piezo1 (e.g., excluding the pore region as used by McHugh et al. [29]) would be required to fully exclude a role for Piezo1 conductance. This would also allow one to address potential roles of Piezo1 ion conductance in endomembranes.

Piezo1 has been shown to localize to focal adhesions and enhance their maturation in a force-dependent manner [57]. It seemed plausible, therefore, that PiCS could be directly or indirectly connected to integrin signaling. To explore this, we activated or inhibited components of the integrin signaling pathway at various levels. Integrin-β1 activation increased the stiffness of control cells, with no further increase by Piezo1 overexpression. This suggests that a common pathway may be at play. This was substantiated by showing that FAK, one of the major downstream effectors of integrin-β1 signaling in response to extracellular stimulation that stabilizes the cytoskeleton [58], is required for PiCS. Since PiCS is not prevented by directly blocking integrin-β1, Piezo1 is unlikely to activate integrin-β1 directly.

ROCK kinases are another group of downstream effectors of integrin-β1, with distinct effects on the actin cytoskeleton: ROCK1 is required for the formation of thick actin fibers, while ROCK2 mediates myosin light chain phosphorylation [59]. In our experiments, inhibition of both ROCK isoforms reduced cell stiffness irrespective of Piezo1 expression, but did not abolish PiCS. Thus, our results suggest that in PiCS, FAK does not activate ROCK, but affects actin assembly more directly, for example by interacting with actin polymerizing proteins [60].

### 4.2. PiCS in HAF and Beyond

HAF share phenotypical similarities with primary cultures of human atrial fibroblasts in terms of collagen deposition, mechanical properties and response to mechanical stimulation [54]. We first confirmed PiCS in HAF and showed that diminished Piezo1 expression leads to decreased cell stiffness, confirming a link between Piezo1 expression level and cell mechanical properties in these cells. Imaging of the actin cytoskeleton indicates that the changes in cell stiffness, induced by alterations of Piezo1 expression, are likely to be related to rearrangements of the actin cytoskeleton. Higher Piezo1 expression levels correlate with higher overall phalloidin intensity, thicker actin bundles, and a higher degree of isotropy. Cell area was not changed in any condition, supporting the notion that Piezo1 does not directly activate/block integrin-β1 [61]. This is a difference compared to observations in epithelial cells where Piezo1 knock-down reduced integrin-β1activation [29].

We had previously shown that mechanical properties of HAF are changed during transforming growth factor-β-induced myofibroblast phenoconversion [36]. Such changes are commonly attributed to enhanced presence of α-smooth muscle actin-positive stress fibers. Here, we show that differential organization of the actin cytoskeleton may be involved as well. Although bundle thickness and isotropy correlate with Piezo1 expression levels, changes seemed rather small to explain the pronounced effects on cell stiffness. It is likely, therefore, that changes in cell mechanical properties during myofibroblast phenoconversion are the result of multiple cytoskeletal remodeling processes, including α-smooth muscle actin, actin, myosin, intermediate filaments, and crosslinkers like filamin A. Further work will be required to disentangle the roles of the various components, both in myofibroblast phenoconversion and in PiCS. Experiments in HAF, including the inhibitors used in our HEK cell experiments, will be valuable to gain more insight in future studies.

### 4.3. Relevance of PiCS in the Context of Altered Environmental Stiffness

HAF respond to higher stiffness of their growth matrix by becoming stiffer themselves, which is in line with the response of primary human atrial fibroblasts [44] and corresponds to one of the canonical cell adaptations to matrix stiffness [62]. We show that Piezo1 contributes critically to this ability, as HAF stiffness adaptation to differences in matrix mechanical properties was abolished with Piezo1 knock-down. Of note, HAF grown on CyPhyGels were stiffer than HAF on plastic, which may be explained by a biphasic relationship between matrix stiffness and cell stiffness, or by differences in surface chemistry (nature and density of ligands for cell attachment) between CyPhyGels and tissue culture plastic. Both states of the matrix used in this study are at the lower end of myocardial tissue (ranging from ~5 to more than 50 kilopascals depending on age and disease state [3,4,5,6,7]), and more work will be required to assess whether Piezo1 knock-down can also impair fibroblast adaptation to stiffnesses that are in the mid or high range of myocardial tissue properties. In addition, it needs to be considered that in vivo cells face a 3D environment, in which mechano-sensing and -adaptation will differ from our 2D model. High-resolution imaging of the actin cytoskeleton of fibroblasts in a 3D environment may help understanding if comparable mechanisms are at play in tissue.

It is unlikely that Piezo1 could overwrite integrin-mediated stiffness sensing, but in the conditions used here, Piezo1 is essential. Our current data does not allow one to discriminate between impaired sensing of, and impaired adaptation to, differences in matrix stiffness. As adaptation is a consequence of sensing, both may have overlapping outcomes. When tissue becomes stiffer for example due to ECM deposition in fibrosis, changes in Piezo1 expression (such as in the context of permanent AF) might impair the way in which fibroblasts sense ECM stiffness, thus preventing adaptation. This might alter their transition to myofibroblasts. Indeed, it has been shown before that stiff growth matrices can trigger myofibroblast phenoconversion [1]. In this context, Piezo1 could be an attractive target to slow or limit the vicious circle in which fibrosis begets fibrosis in cardiac disease.

This positive feedback loop might in parts be explained by the upregulation of IL-6 secretion in response to Piezo1 activation [24]. In our experiments, Piezo1 overexpression resulted in increased IL-6 secretion from HAF. Taking into account that the transfection efficiency for HAF is usually below 5%, the increase in IL-6 we found in the culture medium is remarkable. Elevated IL-6 levels in the culture medium are required for increased cell stiffness in Piezo1-transfected HAF, and for induction of PiCS in neighboring, non-Piezo1-transfected cells, i.e., IL-6 acts in an autocrine and in a paracrine manner. There is some evidence in the literature pointing to a connection between IL-6 and integrin signaling in cardiac fibroblasts, for example in response to differences in matrix stiffness [63]. It remains to be evaluated whether IL-6 signaling itself is sufficient, and integrin signaling is required for paracrine induction of PiCS. Future work will identify the mechanisms that link enhanced IL-6 secretion in HAF to alterations of mechanical properties in remote cells.

## 5. Conclusions

Matrix stiffness sensing allows fibroblasts to adapt to their mechanical environment, affecting a number of cell functions, including myofibroblast phenoconversion and collagen production. Our results identify Piezo1 in HEK cells and HAF as a key component of the cellular matrix stiffness sensing and adjustment responses. We show that PiCS involves integrin signaling and identify an inverse relation between Piezo1 expression levels and F-actin anisotropy. We further establish that PiCS requires autocrine signaling via a process involving increased levels of IL-6 in the culture medium and can additionally be communicated in a paracrine manner to neighboring (non-transfected) cells. The role of Piezo1 as a combined mechano-sensor and -effector requires further investigation, in particular with the view of identifying potential targets for intervention to slow down, stop or prevent fibrosis development.

## Figures and Tables

**Figure 1 cells-10-00663-f001:**
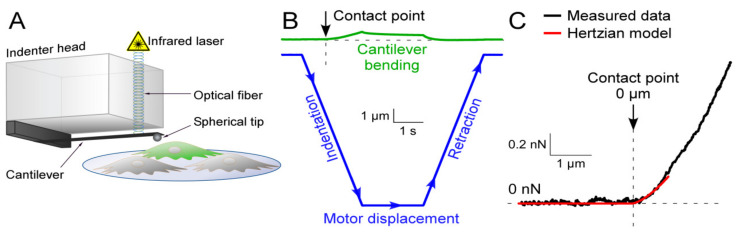
Measuring cell stiffness by nanoindentation: Principle, protocol, and recording. (**A**) nanoindenter used to measure resistance to deformation (stiffness). A spherical glass tip attached to a cantilever of known stiffness is used to indent a sample, e.g., a cell. Cantilever deformation is measured interferometrically and used to calculate the force applied to the sample. Green cell: Transfected cell expressing enhanced green fluorescent protein (EGFP) alone, or EGFP and the gene of interest. (**B**) downward movement of the indenter (motor displacement) beyond the cell surface leads to indentation of the cell and to cantilever bending. (**C**) force vs. indentation curves can be fitted using a Hertzian model for contact mechanics enabling to derive the effective Young’s modulus E_eff_, which is 0.15 kPa in this example.

**Figure 2 cells-10-00663-f002:**
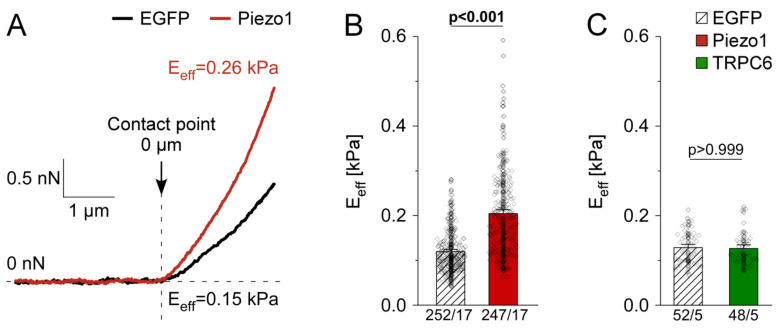
Piezo1 overexpression induces stiffening of human embryonic kidney cell (HEK) cells. (**A**) representative force vs. indentation curves from cells overexpressing EGFP or Piezo1 acquired by nanoindentation. Average stiffness of cells overexpressing EGFP or Piezo1 (**B**) or transient receptor potential channel 6 (TRPC6) (**C**) 3 days post-transfection. n/N = number of cells/ number of experiments.

**Figure 3 cells-10-00663-f003:**
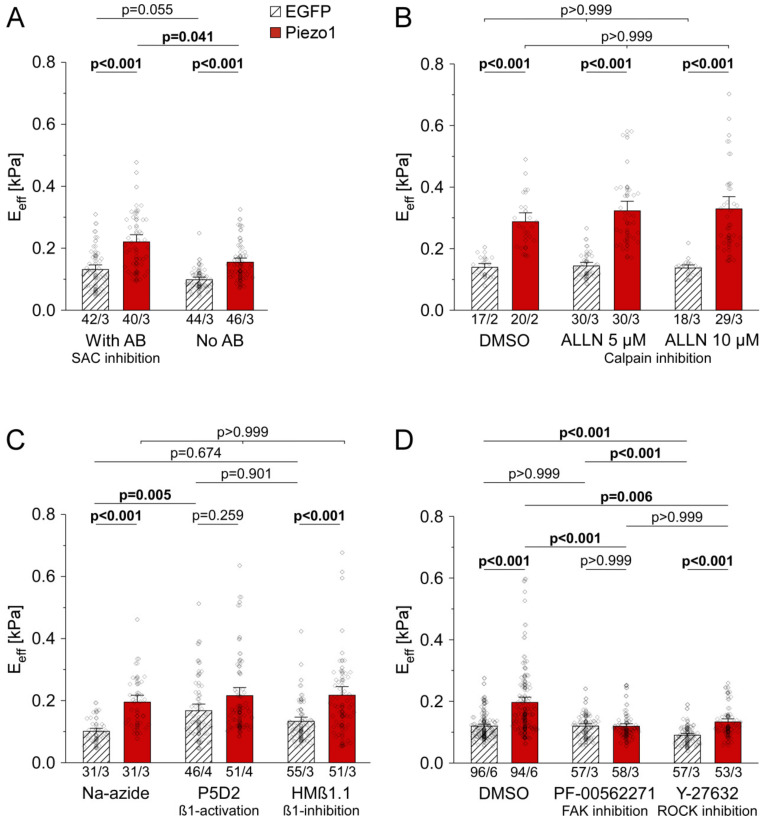
Piezo1-induced stiffening does not require Piezo1 channel activity at the plasma membrane, calpain activity, integrin-ß1 and ROCK, but FAK in HEK cells. Average stiffness of cells: 2 days after wash-out of antibiotics (AB, 172 µM streptomycin and 200 µM penicillin, (**A**)); after DMSO treatment (solvent control, 0.1% *v*/*v*) or different concentrations of the calpain inhibitor ALLN (**B**); after 2 days of treatment with Na azide (solvent control, 0.09% *v*/*v*) or monoclonal antibodies to activate (P5D2, 2.5 µg/mL) or inhibit (HMß1.1, 1.25 µg/mL) integrin-ß1 outside-in signaling (**C**); after 2 days of treatment with DMSO, FAK inhibitor (PF-00562271, 1 µM), or ROCK inhibitor (Y-27632, 1 µM, (**D**)). n/N = number of cells/ number of experiments.

**Figure 4 cells-10-00663-f004:**
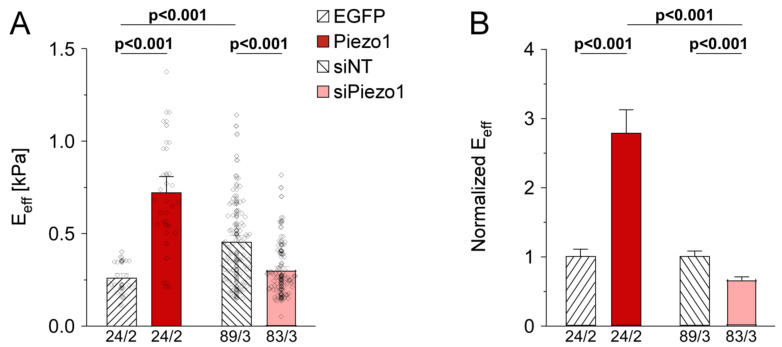
Piezo1-induced stiffening in human atrial fibroblast line (HAF). (**A**) average stiffness of cells 3 days after transfection of expression constructs for overexpression of EGFP or Piezo1, non-targeting siRNA (siNT) or siRNA for knock-down of Piezo1 (siPiezo1). (**B**) data from (**A**) normalized by the mean of the respective control group (EGFP for overexpression, siNT for knock-down). n/N = number of cells/number of experiments.

**Figure 5 cells-10-00663-f005:**
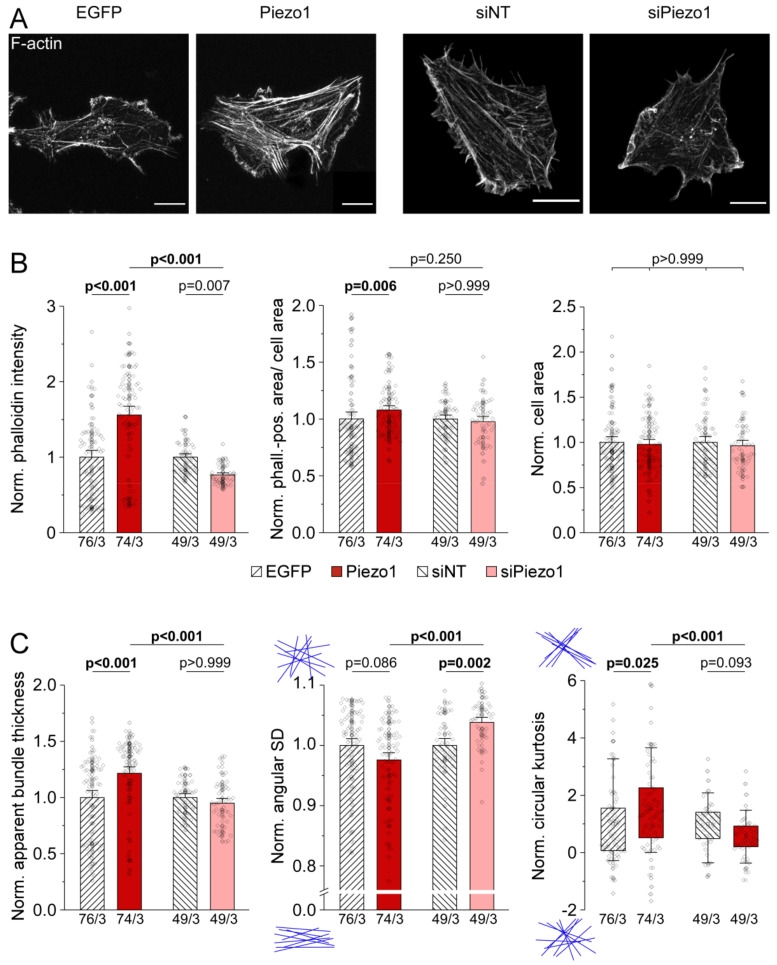
Piezo1-dependent alterations of HAF stiffness are a result of differential organization of the actin cytoskeleton. (**A**) representative images of the actin cytoskeleton of HAF 3 days post-transfection, stained by phalloidin. Scale bars = 20 µm. (**B**) average phalloidin intensity, phalloidin-positive area per cell area, and cell area. (**C**) apparent thickness, angular standard deviation (SD), and circular kurtosis of actin bundles. All data normalized to the mean of the respective control. See Appendix A for raw data. n/N = number of cells/number of experiments.

**Figure 6 cells-10-00663-f006:**
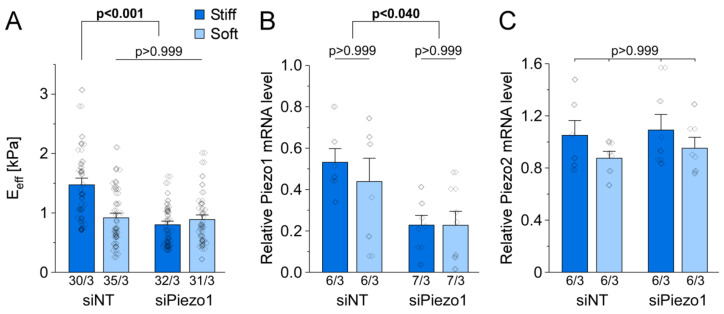
Piezo1 is involved in the adaptation of cell stiffness to matrix stiffness in HAF. (**A**) average stiffness of cells after 4 days of culture on stiff (~6 kPa) or soft (~3 kPa) CyPhyGels. n/N = number of cells/number of experiments. mRNA levels of Piezo1 (**B**) and Piezo2 (**C**) in cells cultured on stiff or soft CyPhyGels 3 days after control intervention or siRNA-mediated Piezo1 knock-down, assessed by qPCR, relative to the expression of glyceraldehyde-3-phosphate dehydrogenase. n/N = number of CyPhyGels/number of experiments.

**Figure 7 cells-10-00663-f007:**
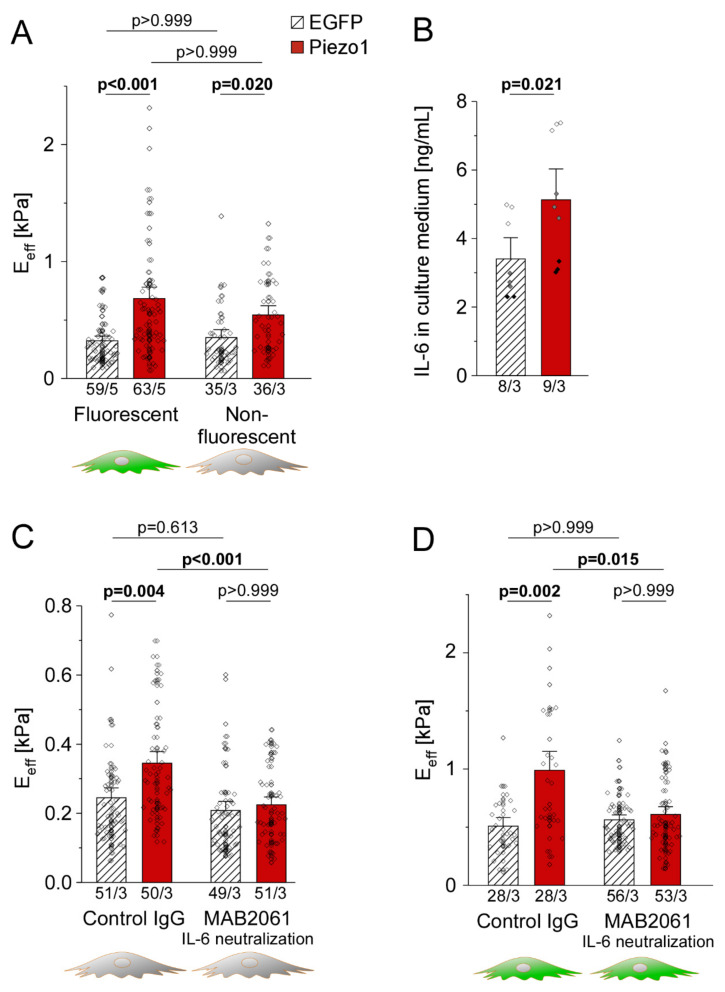
Piezo1-induced stiffening is transmitted to non-transfected cells by a paracrine mechanism dependent on IL-6. (**A**) average stiffness of HAF successfully transfected (fluorescent) or non-transfected (non-fluorescent) in the same dish. (**B**) IL-6 concentration in the culture medium analyzed 3 days after transfection of EGFP or Piezo1. Average stiffness of non-transfected (**C**) and transfected cells (**D**) after treatment with a non-targeting control IgG or IL-6 neutralizing antibody MAB2061 (0.6 µg/mL) for 2 days. n/N = number of cells/ number of experiments (except for B, where n = number of dishes).

**Figure 8 cells-10-00663-f008:**
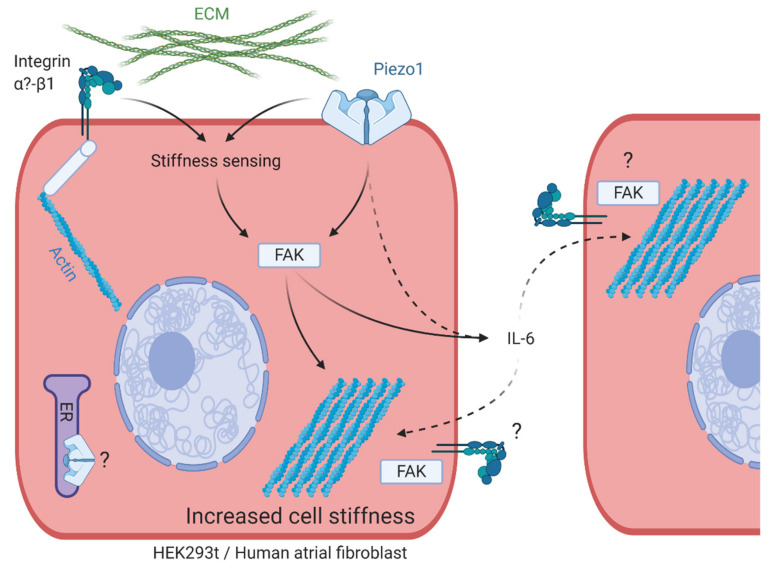
Schematic summarizing the proposed pathway for Piezo1-induced cell stiffening (PiCS). Piezo1 and integrin-β1 can independently activate FAK, leading to a reorganization of the actin cytoskeleton, and ultimately changing the cell’s mechanical properties. Increased Piezo1 expression induces enhanced IL-6 secretion which is required for autocrine and paracrine induction of cell stiffening. ECM = extracellular matrix. FAK = focal adhesion kinase.

**Table 1 cells-10-00663-t001:** Targets, compounds, and concentrations used in this study.

Purpose	Agent	Class	Catalogue No.	Supplier	Final Concentration/Dilution	Solvent (Final Concen- tration)
Calpain inhibition	ALLN	Peptide	208719	Merck Millipore	5 or 10 µM	DMSO (0.05 or 0.1%)
Integrin-β1 activation	P5D2	Mono- clonal antibody	ab24693	Abcam	2.5 µg/mL	Na-azide (0.04%)
Integrin-β1 inhibition	HMβ1.1	Mono- clonal antibody	ab36219	Abcam	1.25 µg/mL	Na-azide (0.09%)
FAK inhibition	PF-00562271	Small molecule	S2672	Selleck Chemicals	1 µM	DMSO (0.1%)
ROCK inhibition	Y-27632	Small molecule	S1049	Selleck Chemicals	1 µM	DMSO (0.1%)
IL-6 neutra- lization	MAB2061	Mono- clonal antibody	MAB2061	R&D Systems	0.6 µg/mL	PBS (0.1%)

ALLN = N-acetyl-leucin-leucin-norleucinal. FAK = focal adhesion kinase. ROCK = Rho-associated kinase. DMSO = dimethyl sulfoxide. IL-6 = interleukin-6. PBS = phosphate buffered saline.

**Table 2 cells-10-00663-t002:** TaqMan assays used for qPCR (all from ThermoFisher, Dreieich, Germany).

Target Gene	TaqMan Assay Number
*PIEZO1*	Hs00270203_m1
*PIEZO2*	Hs00401026_m1
*GAPDH*	Hs02786624_g1

*GAPDH* = Glyceraldehyde-3-phosphate dehydrogenase.

## Data Availability

Data is available from the corresponding author upon request.

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
