# Peer review of "Piezo1 Channels Contribute to the Regulation of Human Atrial Fibroblast Mechanical Properties and Matrix Stiffness Sensing"

_cells, 2021, doi:10.3390/cells10030663_

Round 1
Reviewer 1 Report
The manuscript by Emig et al entitled “Piezo1 channels contribute to the regulation of human atrial fibroblast mechanical properties and matrix stiffness sensing” is an interesting work on the matrix tension sensor channel. Given the physiological relevance for this channel in various fibrotic disorders, the finding presented here is of significance. The manuscript well written with good description of the results, excellent statistical analysis performed for each figure. Listed below are the issues raised by this reviewer. These experimental additions and changes will strengthen the manuscript and improve the confidence in the data presented.
Major issues:
- How did the authors conclude piezo1 overexpression? Without immunostaining or immunoblot experiment, it will be hard to make this claim. There are several good antibodies available to confirm the overexpression by either of these methods. This needs to be performed to show the overexpression of piezo1 in the cells. The data from these experiments should be added to Fig. 1. Also, please provide the cloning details of the plasmid, if it has been used in a previously published manuscript, please provide citation.
- In page 10, where they discuss Y-27632 ROCK inhibitor experiment, please make it clear that the reduction in stiffness observed is “piezo-independent” effect. The sentences do not make this fact very clear.
- Figure 4 requires immunoblots of cells to confirm knockdown of protein piezo1 or immunostained images will suffice.
- The increase in cell stiffness for the non-targeting siRNA is troublesome; please consider using another scramble siRNA. This reviewer will be fine if the authors address all other comments and cannot get this experiment done.
- Figure 6 needs to have measurements from non-treated control cells plated on stiff and soft surfaces. Given that SiNT cells have an increased stiffness to begin with, this control experiment is essential.
Author Response
We thank the reviewer for critical assessment of our manuscript. Please see our detailed replies in the attachment. All changes in the main manuscript were highlighted in the new uploaded version.

Reviewer 2 Report
The manuscript by Emig et. al. investigates the dependence of mechanosensitivity of human atrial fibroblasts on Piezo1. The authors provide data to thoroughly demonstrate the regulation of cellular stiffness through mechanisms downstream of Piezo1. The manuscript is well written with hypothesis tested with under and over expression of Piezo1 and multiple downstream inhibitors to elucidate players in the mechanism. Experiments, sampling, and statistical tests are all well described. While the data support the hypothesis the applicability of this work could be supported examining physiological/pathophysiological ranges or other manners of putting to data into that context. In some cases further analysis could add significant mechanistic understanding. Specific comments are included below.
The substrate stiffness tested in this study are not necessarily connected to the in vivo stiffness experienced. The introduction states that fibrotic or stiff areas of the heart have stiffnesses of ~30kPa while the stiff gels used are only 4.6 kPa. While the results show a sensitivity to stiffness between 2.7 and 4.6 kPa, is this maximized or would it respond further at higher stiffness of 30 kPa? It does not appear that the response is monotonic as cells cultured on the stiff gels had higher cellular stiffness than those grown on rigid substrates in which the stiffness is many orders of magnitude higher. Expansion of the range or further comment on this would be warranted.
The paracrine sensing of stiffness is an intriguing aspect of this work. It would seem that there are experiments that may have already been conducted that would shed further light onto these mechanisms. Particularly in Fig 7C, was the stiffness of any green cells examined in the presence of the IL-6 neutralizing antibody. This would help demonstrate if the cytoskeletal responses were more directly related to Piezo1 or if IL-6 is really the key to elevated cell stiffness. It would require additional experiments, but determining if IL-6 secretion is downstream of substrate stiffness or just Piezo1 overexpression would be informative.
Minor comments:
The manuscript deals in 2D, while fibroblast are mesenchymal cells in a 3D environment. While the mechanosensitive differences between 2D and 3D may not be as distinct as initially stated (Smith et. al. Physiology 2018) a comment on how these results apply to cells in a 3D environment may be warranted.
The methods state some assumptions of the Hertzian model used. One that is commonly used for cells that is generally not fully satisfied is the isotropic substrate.
Fig 3: Many of the inhibitors used here would be expected elicit a change in the actin cytoskeleton. The Image analysis done later in Fig 5 is quite informative. It may be helpful to include it in the context of these inhibitors as well.
Fig 4: The authors state an unexpected shift in the control conditions and make an appropriate shift to normalized stiffness. Overall the sampling robust, however, this unexpected result occurs in the case where sampling is a bit low at N=2.
Results 3.5: “..Piezo1 knock-down resulted in 0.7 times lower fluorescence intensity compared to..” should probably be something similar to “0.7 times of the fluorescence intensity.”
Fig 6: This is related to above, but why is the stiff gel control have higher Eeff while cell on tissue culture plastic are much stiffer? Is there an optimal substrate stiffness for it being translated to cellular stiffness?
Author Response

(The authors gave the same response as above.)

Round 2
Reviewer 1 Report
The authors have responded to all comments. I recommend accepting the manuscript.
Reviewer 2 Report
The manuscript by Emig et. al. has been improved to provide additional detail on paracrine impacts of IL6 and more generally provides further context and applicability to their work. My previous comments have all been adequately addressed. The authors provide a strong account in the literature for the role of Piezo1 in regulating mechanosensitivity of atrial fibroblasts.